# Interactions among Long Non-Coding RNAs and microRNAs Influence Disease Phenotype in Diabetes and Diabetic Kidney Disease

**DOI:** 10.3390/ijms22116027

**Published:** 2021-06-02

**Authors:** Swayam Prakash Srivastava, Julie E. Goodwin, Pratima Tripathi, Keizo Kanasaki, Daisuke Koya

**Affiliations:** 1Department of Pediatrics, Yale University School of Medicine, New Haven, CT 06511, USA; julie.goodwin@yale.edu; 2Vascular Biology and Therapeutics Program, Yale University School of Medicine, New Haven, CT 06511, USA; 3Department of Biochemistry, Dr. Ram Manohar Lohia Institute of Medical Sciences, Lucknow 226010, India; pratimatripathi.lko@gmail.com; 4Internal Medicine 1, Shimane University Faculty of Medicine, Izumo 693-0021, Japan; kkanasak@med.shimane-u.ac.jp; 5Department of Diabetology and Endocrinology, Kanazawa Medical University, Ishikawa 920-0293, Japan

**Keywords:** long noncoding RNAs, microRNAs in kidney, kidney fibrosis, EMT, EndMT, diabetes mellitus, diabetic kidney disease

## Abstract

Large-scale RNA sequencing and genome-wide profiling data revealed the identification of a heterogeneous group of noncoding RNAs, known as long noncoding RNAs (lncRNAs). These lncRNAs play central roles in health and disease processes in diabetes and cancer. The critical association between aberrant expression of lncRNAs in diabetes and diabetic kidney disease have been reported. LncRNAs regulate diverse targets and can function as sponges for regulatory microRNAs, which influence disease phenotype in the kidneys. Importantly, lncRNAs and microRNAs may regulate bidirectional or crosstalk mechanisms, which need to be further investigated. These studies offer the novel possibility that lncRNAs may be used as potential therapeutic targets for diabetes and diabetic kidney diseases. Here, we discuss the functions and mechanisms of actions of lncRNAs, and their crosstalk interactions with microRNAs, which provide insight and promise as therapeutic targets, emphasizing their role in the pathogenesis of diabetes and diabetic kidney disease

## 1. Long Non-Coding RNA (lncRNA)

Long non-coding RNAs (LncRNAs) account for the major class of the noncoding RNAs in the genome and are linear transcripts longer than 200 nucleotide sequences that share similar characteristics with mRNAs [1]. Most LncRNAs are transcribed by RNA polymerase II and have capping at the 5′ end and splicing and the polyadenylated tail at the 3′ end. LncRNAs have defined promoter regions [1]. However, compared to mRNA, lncRNAs do not have open reading frames (ORFs) and have lesser exons (lncRNAs contain around 2.8 exons whereas mRNA contains 11 exons). LncRNAs can be transcribed as a whole or partial natural antisense transcripts (NAT) to coding genes, or located between genes or within introns [1]. Some lncRNAs originate from pseudogenes [2]. LncRNAs can be divided into several subtypes according to their position (such as antisense, intergenic, overlapping, intronic, bidirectional and recessed) and transcriptional direction in relation to other genes [3,4].

## 2. Synthesis Procedure and Location

Gene expression profiling and in situ hybridization studies have shown that the expression of lncRNAs can be tissue- and cell-specific, and may vary spatially, temporally or in response to stimuli [5]. Many lncRNAs are located exclusively in the nucleus, however, some are cytoplasmic or are located in both nucleus and cytoplasm. LncRNAs are critical regulators of gene expression and have functions in a wide range of cellular and developmental processes [5]. LncRNAs function through both inhibition and activation of genes [6].

LncRNAs are classified into four groups based on their location in the genome: (1) the intergenic lncRNAs, (2) the sense or antisense lncRNAs, (3) the intronic lncRNAs and (4) the processed transcripts; these lncRNAs reside in a gene-loci that has no ORF [6,7].

Based on their functions, lncRNAs have been characterized as signal, decoy, scaffold, guide, enhancer RNAs and short peptides [8,9]. Signal lncRNA acts as a molecular signal that regulates transcription processes [10]. Decoy lncRNAs act by reducing the availability of key molecules that are involved in gene regulation. These lncRNAs alter the transcription level by sequestering regulatory factors, and microRNAs, hence minimizing their expression level [11]. The scaffold class of lncRNAs provides structural support for complex proteins [12] and transcriptional activation or repression is conferred depending on the types of regulatory proteins and RNAs that exist [13]. Guide lncRNAs interact with ribonucleoproteins complex and influence the gene transcriptional level [14].

## 3. LNCs in Diabetic Kidney Disease

Available evidence has indicated the important roles of lncRNAs in the pathophysiology of diabetic kidney disease (DKD), and the crosstalk between lncRNAs and DKD were reported in recent years [15,16,17,18,19]. Altered expression levels of lncRNAs play key roles in the development of proteinuria and associated diabetic nephropathy (DN) [15,20]. LncRNAs are involved in the progression of kidney disease through regulation of many important factors, such as pathologic processes in mesangial cells, podocytes, reactive oxidative species, epithelial-to-mesenchymal transition (EMT), endothelial-to-mesenchymal transitions (EndMT) and actions on microRNAs [21,22,23]. 

Several lncRNAs participate in the regulation of renal disease (Table 1). For example, plasmacytoma variant translocation (PVT1) participates in the development of DN by regulating ECM accumulation. PVTI is the first non-coding RNA reported to be associated with kidney disease, which is highly expressed in human renal mesangial cells under high-glucose conditions and significantly promotes the expression of fibronectin protein, type IV collagen, TGF-β1 and type I plasminogen activator inhibitor [20,24,25]. Metastasis-associated lung adenocarcinoma transcript 1 (MALAT1) is aberrantly upregulated in early DN [26,27,28]. MALAT1 initiates inflammation and oxidative stress; these pathogenic pathways regulate glucose-stimulated induction of the proinflammatory cytokines IL-6 and TNF-α by activating serum amyloid antigen 3. These changes alter endothelial cell stability in DN [20,29]. Gm4419 is located in chromosome 12 and is a regulator of the nuclear factor kappa light-chain enhancer of activated B cells (NF-κB), which is a crucial inflammatory factor for DN [20,30]. Gm4419 interacts with p50 and induces the NF-κB/NLRP3 inflammasome signal transduction pathway in mesangial cells, which is associated with inflammation, fibrosis and proliferation in high-glucose conditions [30]. NR_033515 is significantly upregulated in serum of DN patients [31]. Overexpression of NR_033515 promotes mesangial cell proliferation and inhibits apoptosis [31]. NR_033515 has been shown to upregulate the gene expression levels of proliferation-related genes, fibrosis-associated genes and EMT markers by targeting miR-743b-5p [31]. Kidney-specific deletion of Erbb4-IR has been shown to confer protective effects against DN complications [32]. Erbb4-IR inhibits the expression level of reno-protective miR-29b. Therefore, the level of fibrosis was enhanced by Erbb4-IR i diabetic kidneys [32]. Antisense mitochondrial noncoding RNA-2 (ASncmtRNA-2) is a mitochondrial lncRNA [33]. ASncmtRNA-2 is upregulated in ageing and senescence in endothelial cells [33]. ASncmtRNA-2 induces oxidative stress and causes tubular injury through (i) accelerated lipid peroxidation and protein crosslinking, (ii) damage to DNA and (iii) promoting inflammatory pathways, such as NF-κB and transforming growth factor-β1 (TGFβ1) [33]. Lnc-MGC is regulated by an ER stress-related transcription factor, CHOP (C/EBP homologous protein), and by TGFβ1-dependent and independent mechanisms [34]. ER stress increases in patients with progressive DN [34]. Nuclear enriched abundant transcript-1 (NEAT1) is highly expressed in high-glucose conditions, and interacts with AKT/mTOR pathways [35,36]. NEAT1 inhibition leads to suppression in levels of TGFβ1, FN and COL4A1 in DN [36]. NEAT1 promotes high glucose-stimulated mesangial cell hypertrophy by targeting the miR-222-3p/CDKN1B axis [37]. Similarly, lncRNA ERBB4-IR is involved in the development of renal fibrosis in diabetes and its silencing in diabetic mice protects against albuminuria and fibrogenic processes [32,38].

Conversely, the expression of CYP4B1-PS1-001, which upregulates the nucleolin protein level, is suppressed in high-glucose conditions [39,40]. CYP4B1-PS1-001 overexpression suppresses the levels of FN, COL4A1 and proliferation markers in diabetic mice [40]. Another example of a reno-protective lncRNA is the lncRNA ENSMUST00000147869, which target ECM production in kidneys of diabetic mice [41]. ENSMUST00000147869 affects ECM synthesis and dramatically decreases the levels of fibronectin and collagen IV in mesangial cells under high-glucose conditions [41], though the exact role of this lncRNA is unknown. TUG1 functions as a repressor of miR-377. miR-377 directly targets the 3′UTR of PGC-1α and fibrosis markers. Therefore, TUG1 upregulates the level of PGC-1α and alleviates ECM production and downregulates the expression levels of proinflammatory cytokines in high-glucose stimulated mesangial cells [42]. Myocardial infarction-associated transcript (MIAT), also known as retinal noncoding RNA 2 (RNCR2), has been known to associate with myocardial infarction [35]. MIAT regulates cell viability through stabilizing nuclear factor erythroid 2-related factor 2 (NRF2) expression in renal tubules [20]. NRF2 pathologically and functionally protects the kidneys against diabetic damage [43]. Interestingly, expression of Nrf2 can be enhanced by MIAT overexpression in glucose-treated renal tubular epithelial cell lines [44]. Cancer susceptibility candidate 2 (CASC2) has critical functions in tumorigenesis [45]. Downregulated expression of CASC2 has been observed in serum and kidney tissues in diabetic kidneys and is predictive of diabetic complications [46]. Low-plasma levels of CASC2 are associated with higher risk of renal failure in DN patients [47,48]. Another lncRNA, 1700020I14Rik, which is located in chromosome 2 (Chr2: 119594296–119600744), functions as an endogenous RNA and regulates the expression levels of microRNAs in diabetes [20,49]. Overexpression of 1700020I14Rik suppresses the expression level of miR-34a-5p by Sirt1/HIF-1α signal pathway and accelerates fibrosis in mesangial cells [49]. CYP4B1-PS1-001 is downregulated in early DN [40]. Its overexpression inhibits fibrosis of mesangial cells by interacting with nucleolin [40]. Gm15645 is downregulated in DN and high-glucose-stimulated, cultured podocytes [50]. The mechanism of Gm15645 is contrary to that of Gm5524, which affects podocyte cell death and autophagy regulation in DN [50]. LINC01619 regulates miR-27a/FoxO1 (forkhead box protein O1) and ER stress-associated podocyte cell injury in diabetes [51]. Downregulated expression levels of LINC01619 are associated with proteinuria and declines in kidney function in DN patients; hence, targeting LINC01619 is one of the potential therapeutic options for treatment of DN [51]. Figure 1 demonstrates the lncRNAs involvement in influencing EMT, EndMT and glomerular injury in diabetic nephropathy.

## 4. LncRNAs Involvement in the Regulation of EMT

EMT involves a series of processes by which epithelial cells lose their epithelial characteristics and acquire properties of mesenchymal cells [52,53,54,55,56,57]. Figure 1 depicts the involvement of lncRNAs in the regulation of EMT, EndMT and mesenchymal cells. Epithelial cells are normally associated tightly with their neighbor cells. In contrast, mesenchymal cells do not form intercellular adhesion complexes [58]. Mesenchymal cells are elongated in shape and exhibit end-to-end polarity and focal adhesions, allowing for increased migratory capacity [58]. The main function of fibroblasts, which are prototypical mesenchymal cells that are found in several tissues, is to maintain structural integrity by secreting extracellular matrix (ECM). Fibroblast-specific protein 1 (FSP-1), alpha-smooth muscle actin (SMA), vimentin, fibronectin and collagen I are the markers that characterize mesenchymal products in diabetic kidneys [58,59,60]. Inflammation results in the recruitment of multiple types of cells that are involved in the induction of EMT processes. Elevated levels of TGFβ1, platelet-derived growth factor (PDGF), epidermal growth factor (EGF) and fibroblast growth factor-2 (FGF-2) contribute to EMT processes [59,60,61]. MALAT1, NR_033515, Erbb4-IR, GAS5 and CJ241444 are involved in tubular injury and contribute to EMT processes whereas MIAT and LncRIAN have shown tubular protective activity and may have regulate EMT processes in diabetic kidneys (Figure 1).

## 5. LncRNAs Involvement in the Regulation of EndMT

Endothelial cells form fibroblasts by undergoing a transition, referred as EndMT [57,58,62,63,64,65]. EndMT is characterized by the loss of endothelial cell phenotypes, and gain of mesenchymal proteins [58,62,64,65,66,67]. EndMT participates in fibrogenic processes in kidneys and, in diabetic kidneys, can alter the physiology and function of other neighboring cells [58,62,65,68]. Pathological stimuli such as inflammation, diabetes and ageing influence EndMT events in the kidneys [69]. Endothelial SIRT3, the nuclear receptor glucocorticoid receptor (GR) and cell surface FGFR1 are critical regulators of TGFβ signaling and EndMT in diabetic kidneys [70,71,72,73]. The kidneys of diabetic mice showed both progressive glomerular sclerosis and tubulointerstitial fibrosis, which was associated with approximately 40% of all FSP-1-positive cells and 50% of αSMA-positive stromal cells were CD31-positive [74]. Similarly, in the kidneys of COL4A3 knockout mice, 45% of all αSMA-positive fibroblasts and 60% of all FSP-1-positive fibroblasts were CD31-positive, suggesting that these fibroblasts are of endothelial origin and that EndMT might contribute critically to the development and progression of renal fibrosis [74]. During the process of EndMT, biochemical changes lead to the decreased expression of endothelial markers and the gain of mesenchymal markers such as FSP-1, αSMA, smooth muscle 22-alpha (SM22α), N-cadherin, fibronectin, vimentin, types I and III collagen, nestin, cluster of differentiation, 73 (CD73), matrix metalloproteinase-2 (MMP-2) and matrix metalloproteinase-9 (MMP-9) [58,75,76]. MALAT1, Erbb4-IR and ASncmtRNA2 cause endothelial cell injury and may involve EndMT-associated renal fibrosis (Figure 1). LncRNA H19 is associated with kidney fibrosis by activating EndMT processes in diabetes (Figure 1).

## 6. LNCs Interaction with microRNA

The miRNA and lncRNA interaction is one of mechanisms for regulation of gene expression [77]. This multilevel regulation is involved in almost all the physiological and cellular processes at the transcriptional, post-transcriptional and post-translational levels [77,78]. In some studies, it has been reported that miRNA triggers lncRNA decay [77]. On the contrary, lncRNAs generate miRNAs, act as miRNA sponges and miRNA decoys, and compete with miRNA for binding at mRNAs [77].

LncRNA genes can harbor microRNAs and these microRNAs can be released by post-transcriptional processing. For example, the lncRNA PVT1 serves as a host of miR-1207-5p and has been implicated in DN [79]. microRNAs are often present in clusters, having been localized to the PVT1 locus, and are upregulated by high-glucose and affect extracellular matrix accumulation [80]. MiRNA clusters in lncRNAs can get very large as demonstrated by a megacluster of more than 40 miRNAs harbored in lnc-MGC [34]. This cluster is induced in diabetic glomeruli through endoplasmic reticulum stress signaling, which responds to high glucose and TGFβ-activation as well [34].

The interactions between microRNAs and lncRNAs are important to study the key steps in DN progression. DN mice exhibit interactions between lncRNA CJ241444-miR-192 that induces TGFβ1/Smad3 signaling [81] and lncRNA Erbb4- IR-miR-129b activates collagen genes and ECM genes and, hence, kidney injury [82]. These lncRNAs can act as miRNA sponges [32,81]. Similarly, lncRNA PVT-1 participates in ECM accumulation via the actions of its-derived miRNAs, miR-1207-5p and miR-1207-3p [25]. Under high-glucose conditions, higher expression of both PVT-1 and its miRNAs cause an increase of TGFβ1/Smad3 signaling and ECM accumulation [25]. Similarly, miR-379 clusters that are regulated by ER stress in DN and lncMGC are also hosted in this same cluster [34]. LncMGC regulates the expression of the miR-379 clusters, and the upregulation of the miR-379 clusters induces ECM accumulation and renal hypertrophy [34]. Thus, antagonism of lncMGC expression can be used as a potential therapeutic for DN to reduce the effects of the miR-379 cluster, following ER stress [34]. Besides that, lncRNA NEAT1 antagonism is also a potential therapy, since NEAT1 antagonism leads to the suppression of ECM deposition via the reduction of ASK1, FN and TGFβ1 production [83]. This NEAT1-associated ECM suppression is due to its interaction with miR-27b-3p, and its target, the TGFβ and Zeb1 [83]. Administration of the antiapoptotic lncRNA, TUG-1, suppresses miR-377 expression and its target gene PPARγ and thus prevents ECM accumulation in DN mice [42]. Therefore, treatment to increase TUG-1 expression could be beneficial to treat the DN phenotype and restore kidney structure, although further studies are needed to understand its potential [42]. These findings will allow the development of an understanding of the interactions between lncRNAs and their target miRNAs, that can be useful for therapeutic target selection to prevent ECM deposition and for the management of DN progression. Figure 2 demonstrates LncRNAs and microRNAs interactions in the regulation of diabetic nephropathy.

## 7. LNCs in the Regulations of Antifibrotic microRNAs Crosstalk

TGFβ suppresses antifibrotic miRNAs such as miR-29 clusters and miR-let-7 clusters [84]. Suppression of such TGFβ1-regulated crosstalk of miRNAs was reported in type I diabetic subjects who had higher rates of ESRD progression [85]. The data from our laboratory reveal that clusters of the miR-29 family and the miR-let-7 family showed a protective effect against endothelial-to-mesenchymal transition (EndMT) and demonstrate bidirectional regulation under physiological conditions [86,87,88,89]. This bidirectional regulation is essential for endothelial cell homeostasis and protects against EndMT in diabetic kidneys [76]. Targeting EndMT is one of the potential therapeutic options for the treatment of diabetic kidney fibrosis [56,58]. miR-29 clusters show negative, bidirectional regulation with TGFβ receptors [76]. miRNAs regulate gene expression of each other directly or indirectly. This crosstalk phenomenon is linked with maintenance of antifibrotic activity in the kidney and its disruption results in accelerated renal fibrosis [76]. Interventions that prevent the disruption of this crosstalk are beneficial in protecting against kidney diseases [56,86]. DPP-4 inhibition shows suppression in TGFβ signaling-driven EndMT in diabetic kidneys by elevating miR-29 clusters [67,88]. miR-29 clusters target the profibrotic molecule DPP-4, and its inhibition elevates the miR-29 level; therefore, DPP-4 inhibitors are potential leads for the treatment of diabetic nephropathy [88].

MiR-let-7 inhibits TGFβ receptor 1 [90], and TGFβ-smad3 signaling has been demonstrated as an inhibitory pathway for miR-29 gene expression [84,88,91,92]; therefore, as expected, miR-let-7 induces the expression of miR-29 in endothelial cells. An alternative mechanism of miR-29-linked-miR-let-7 expression was explained by the interferon-gamma (IFNγ)-FGFR1 axis. miR-29 targets IFN-γ [93] and moreover, IFN-γ inhibits FGFR1. FGFR1 plays crucial roles in the expression of miR-let-7 family clusters [90]. Downregulation of miR-29 clusters causes a rise in IFN-γ levels, which subsequently discourage FGFR1 and FGFR1-associated expression of miR-let-7 clusters. This suppression of miR-let-7 expression causes activation of TGFβR1 protein expression. Triggering TGF-β/smad3 signaling in turn inhibits the expression of miR-29 family clusters [88].

AcSDKP is a key peptide that is partially synthesized in the distal tubular regions from the enzymatic action of polyoligopeptidase on thymosinβ4 and is degraded by angiotensin converting enzyme. Hence, angiotensin converting enzyme inhibitors have been shown to elevate the level of AcSDKP in the plasma of mice and diabetic subjects [86,89]. Several studies have been analyzed for renal protective abilities of AcSDKP and ACE inhibitors can perform antifibrotic activities by partially elevating AcSDKP levels [70,89,94]. Most importantly AcSDKP is a key endogenous peptide that restores kidney structure and suppresses renal fibrosis by counteracting DPP-4-associated EndMT through elevating microRNA crosstalk regulations between miR-29 and miR-let-7 [86]. Moreover, inhibition of ACE elevates the level of AcSDKP and cause upregulation of antifibrotic microRNAs and restores the antifibrotic cross-talk in cultured endothelial cells, while angiotensin receptor blockers have minimal effects [76,86,89]. These events control crosstalk regulation between miR-29s and miR-let-7s in fibrotic kidneys of diabetic mice [86]. AcSDKP maintains kidney homeostasis partly by elevating the bidirectional regulation between miR-29s and miR-let-7s [76,86].

Lnc-H19 expression is upregulated in TGFβ2-induced endothelial cells and in fibrotic kidneys of diabetic mice [22]. H19 suppression significantly reduces EndMT and kidney fibrosis [22]. The upregulated H19 expression in diabetic kidneys is associated with downregulated levels of miR-29a [22]. H19 and miR-29 association contributes to a regulatory network involved in EndMT [22]. Similar H19 regulatory mechanisms have previously been reported, such as the H19/miR675 pathway, which inhibits cell growth and Igf1r expression [95]; H19/Let-7-mediated inhibition of HMGA2-mediated epithelial-to-mesenchymal transition [96] and the H19/miR-675 axis inhibits prostate cancer metastasis via TGFβ1 [97]. Xie et al. (2016) also found that H19 interaction with miR17 contributed to a regulatory network involved in renal fibrosis [98]. H19 acts as a competitive endogenous RNA. The regulatory network integrates the transcriptional and post-transcriptional regulatory network of EndMT and renal fibrosis [22]. Interestingly, inhibition of H19 only altered miR-29a levels, not miR-29b or miR-29-c, and suppressed TGF-β/Smad signaling in order to regulate EndMT and renal fibrosis in diabetes [22].

Figure 3 demonstrates the involvement of LncRNAs in the regulation of microRNAs crosstalk and its implications in the mesenchymal activation in diabetic kidneys.

## 8. LncRNA-miRNA-Based Treatment for DKD, Future directions and Perspectives

Many non-coding RNAs (miRNAs, lncRNAs and circRNAs) regulate the expression of critical genes involved in DN phenotypes. These non-coding RNAs (nc RNAs) are stable in biological fluids and can offer potential biomarkers in a diverse array of diseases. Non-coding RNAs are involved in the disease processes of hypertrophy, ECM synthesis, apoptosis and renal fibrosis. Moreover, some research has advanced to synthesize ncRNAs-based treatments, with a few of these ncRNAs already in the clinical trial phase. Therefore, these ncRNAs-based therapeutics would be an alternative approach for the treatment of DN [99].

MiRNA-based therapeutics can be used as alternative therapy for treatment of several diseases including diabetic nephropathy. The application of artificially synthesized oligonucleotides to mimics (miRNA mimics) or knockdown microRNAs (antagomiRs) has evolved [99,100]. In this series, locked nucleic acid (LNA) inhibitor was developed to suppress a specific miRNA expression or action [99,100]. Dramatically, LNA-miR-192 treatment improves the DN phenotype, and thus can be utilized as a potential DN therapy [101].

Other work has showed that subcutaneous injection of anti-miR-21 suppressed the fibrosis level in chronic kidney disease mice [102]. miR-29 family significantly improves renal structure and fibrosis in DN mice [103], thus anti-miR-29-based therapy could be potentially used as an alternative option for DN treatment. miRNA-based treatment is gaining momentum over the last decade. However, the problem lies within the delivery methods. miRNAs regulate several targets at the same time; thus, they may affect other pathways. Therefore, research in miRNA-based therapies is now switching to focus on delivery methods and efficacy and safety to target a specific route and tissue localization [104,105,106].

Moreover, the size of the therapeutic molecule should be small enough to cross the endothelium to the organ or site of interest and should not be filtered out by the kidney [107]. Interestingly, this filtration problem is an advantage for miRNA-based treatment, since the epithelial cells reabsorb the therapeutic agents from the ultrafiltrate, thus reducing the loss [107,108]. Therefore, it is believed that miRNA-based treatments could be safely used for DN subjects, although advanced work or large clinical trials are still needed. Several miRNA-based treatments have advanced to clinical trials, though none for the treatment of DN. Miravirsen (LNA based miR-122 inhibitor) already entered phase II clinical trials to treat HCV infection in patients [109]. Many miRNA-based therapeutics are currently in development for several other diseases; therefore, the use of miRNA-based treatment for DN is a new hope. Another possible treatment option is lncRNAs-mediated treatment for DN. It is comparatively favorable to targeting the lncRNA expression when compared to miRNAs, because of its functional role in transcriptional control, tissue-specific expression and disease-specific alterations. LncRNAs are mainly present in the nucleus; synthetic antisense oligonucleotides (ASOs) are widely suggested to silence the lncRNA expression in the nucleus by commencing RNase H-dependent degradation [110,111]. Design of ASOs is very important as it should bind to the LncRNA-specific site and target a single lncRNA. Furthermore, the real challenge is to treat with ASO in vivo. Similar to miRNA-based treatments, the problems lie within the delivery efficiency and efficacy. Another problem related to lncRNAs based therapeutics is the heterogeneous nature and unconserved intron sequence of lncRNAs [1,112]. Further studies are needed to identify small molecules that induce the expression of renal protective non-coding RNAs. There is a need to search for compounds that induce the expression of antifibrotic non-coding RNAs in diabetic kidneys, such as flavonoids, chalcones, polyhydroquinolines, propiophenone derivatives, deoxyandrographolides, 2-methoxy-estradiol and thiazolidin-4-one derivatives; these synthetic or plant-based compounds have shown protective effects in mouse models of diabetes mellitus [113,114,115,116,117,118,119,120,121,122,123,124,125], and could be further tested and used in the management of DN. ncRNAs play crucial roles in the pathogenesis of type II DM and diabetic complications; despite their limitations, tissue-specific microRNAs expression should be further studied [56,126,127]. Physiological dysfunction, metabolic alterations, ER-stress and inflammation are observed before later features such as proteinuria, which is a major contributor to the development of DKD [20]. Proteinuria determines the cardio-renal outcomes of patients with DKD [128,129,130]. Higher proteinuria leads to tubular damage and is associated with renal inflammation and interstitial fibrosis in diabetes [129,130,131]. Minutolo et al. studied the crucial role of proteinuria in patients who have chronic diabetic kidney disease (DM-CKD), and discussed new information on cardio-renal prognosis in DM-CKD patients [128]. In the absence of proteinuria, DM-CKD patients did not have increased cardio-renal risk when compared with non-diabetic CKD patients [128]. However, in CKD patients with proteinuria, the risk of end-stage renal disease was mainly due to the proteinuria level independent of diabetes [20,132]. The physiological and cellular roles of altered sets of microRNAs and lncRNAs are relevant to study proteinuria and associated DN. In addition, lncRNAs such as GAS5 and GM6135, which are upregulated during renal inflammation, might be addressed by a Lnc-inhibitor [133,134]. Similarly, research on circular RNAs and their role in the health and disease of diabetic kidneys is gaining momentum as well. circRNA_15698, circLRP6, circACTR2, circHIPK3 and circ_0000491 are associated with renal inflammation and fibrosis whereas circRNA_010383 is reno-protective [135,136,137,138,139,140]. Therefore, better understanding of the role of these regulatory circular RNAs in the physiology of diverse kidney cell types is needed. Table 1 presents the list of lncRNAs and circular RNAs, and their targets in kidney disease. 

The role of lncRNAs should be analyzed in preclinical settings before utilizing their therapeutic potential in the management of diabetic nephropathy. Hence, extensive researches demonstrating the role of miRNAs and LncRNAs interaction are needed to validate the possibility of using these miRNAs/lncRNAs-based treatments in proteinuria and associated DN.

## 9. Conclusions

miRNAs and lncRNAs interactions influence DKD progression by targeting genes related to fibrogenesis, ER stress, inflammation, oxidative stress and metabolic dysfunction [8,49,110]. Identification of pathways regulating early-stage (physiological dysfunction, metabolic alteration, ER-stress and inflammation) and late-stage (proteinuria) features are of key importance in studies of DN pathogenesis. miRNAs and LncRNAs interactions open a wide area for basic research and for the development of new therapeutic options against diabetic complications including DKD.

**Table 1 ijms-22-06027-t001:** LncRNAs and circular RNAs in diabetes and diabetic kidney disease.

lncRNAs	Expression	Samples	Targets	Functions	References
Plasmacytoma variant translocation (PVT1)	Up	High-glucose stimulated mesangial cells	Fibronectin, collagen IV, TGF-β1, and PAI-1	DN, ECM accumulation.	[20,24,25]
Metastasis associated lung adenocarcinoma transcript 1 (MALAT-1)	Up	Endothelial cells, STZ mice Podocytes, HEK-293 cells Renal tissues, proximal tubular epithelial cells Serum	IL-6, TNF-α, SAA3, miR-29b CTNNBIP1, SRSF1 miR-23c, ELAVL1, NLRP3 miR-499a	renal fibrosis, disrupts endothelial cell stability Podocytes cell damage Injuries in tubular cells DN phenotypes	[26,27,28,29]
Gm4419	Up	High-glucose stimulated mesangial cells	NF-κB/NLRP3	Fibrosis, cell proliferation	[30]
GM5524	Up	Diabetic tissues, High-glucose stimulated podocytes	Bcl2 and Bax protein LC3/ATG autophagy pathway	DN, Podocytes cells damage	[50,51]
NR_033515	Up	Serum, HEK293 T cells, mesangial cells	PCNA, cyclin D1, P38, ASK1, fibronectin, and α-SMA, E-cadherin and vimentin and miR-743b-5p	DN phenotypes, EMT and cell proliferation	[31]
Erbb4-IR	Up	Renal tissue	miR-29b, TGF-β/Smad3	Renal fibrosis	[32,38]
Antisense mitochondrial noncoding RNA-2 (ASncmtRNA-2)	Up	Endothelial cells	ROS, (i) inducing lipid peroxidation, protein crosslinking, and the formation of DNA adducts; (ii) inducing direct damage to cellular DNA; and (iii) activating multiple cellular signaling pathways, including NF-κB and TGF-β1.	Damage to endothelial cells, Ageing, replicative senescence and fibrosis	[33]
Lnc-MGC	Up	Renal tissues Podocytes Mesangial cells	Endoplasmic reticulum (ER) stress-related transcription factor, CHOP (C/EBP homologous protein), TGF-β1.	ER stress, renal fibrosis, glomerular hypertrophy, and podocyte cells injury EMT and DN.	[34]
GAS5	Up	Human tubular epithelial cells	miR-27, P53, CASP3, NF-κB, BNIP3	Tubular cell apoptosis	[133]
GM6135	Up	Glucose-stimulated-mesangial cells	TLR4, miR-203	Renal inflammation and fibrosis	[134]
LnC-H19	Up	Diabetic mice UUO mice Endothelial cells	TGF-β/Smad3, miR-29a	Renal inflammation and fibrosis	[22,96,97,98,99]
CJ241444- miR-192	Up	Renal cortex and mesangial cells	TGF-β, Akt, Col1a2, Col4A1, Smad, Ets1, miR-192	Glomerular fibrosis	[81]
NEAT1	Up	Renal tissues	Akt, Mtor, collagen IV, Fibronectin, TGF-β1. Zeb1, miR-27b-3p, Ask1, fibronectin	Glomerular fibrosis Mesangial cell proliferation	[36,37]
Circular noncoding RNAs circRNA_15698	Up	Renal cortical cells mesangial cells	collagen IV, collagen I, Fibronectin, TGF-β1. miR-185	Renal fibrosis	[135]
circLRP6	Up	Renal cells	miR-205, HMGB1	DN progression	[136]
circACTR2	Up	Tubular cells	interleukin (IL)-1β, collagen IV and fibronectin	Pyroptosis, Fibrosis in Renal Tubular Cells	[137]
circHIPK3	Up	DN tissues, Glucose-stimulated mesangial cells	Cyclin D1, PCNA, TGF-β1, Collagen I, Fibronectin and miR-185	DN progression	[138]
circ_0000491	Up	Glucose-stimulated-mesangial cells	TGFβR1, miR-101b	Glomerular fibrosis	[139]
circRNA_010383	down	Kidneys of db/db mice mesangial cells	Sponges for miR-135a	DN	[140]
Taurine up-regulated 1 (TUG1)	down	Glucose-stimulated mesangial cells, renal cortex, mesangial cells	endogenous sponge of miR-377, PGC-1α, PAI-1, TGF-β1, FN, collagen IV	Mesangial cells damage, podocyte cell death	[42]
Myocardial infarction-associated transcript (MIAT)	down	HK-2 cells	Nuclear factor erythroid 2-related factor 2 (Nrf2), Acta2	tubular cells damage	[20,35]
Cancer susceptibility candidate 2 (CASC2)	down	serum and renal tissues	JNK pathway	renal failure, podocyte cell death	[45,46,47,48]
ENSMUST00000147869	down	mesangial cells, renal cortex	ECM synthesis, fibronectin and Collagen IV	Mesangial cells damage	[41]
1700020I14Rik	down	mesangial cells, Renal tissues	miR-34a-5p, Sirt1, HIF-1α	renal fibrosis	[49]
CYP4B1-PS1-001	down	mesangial cells, renal tissues	nucleolin (NCL), ubiquitin proteasome-dependent pathway	mesangial cells proliferation and fibrosis	[39,40]
Gm15645	down	Kidneys of Db/db mice and high-glucose-stimulated podocytes	Bcl2/Bax and LC3/ATG pathways	DN, podocyte cell apoptosis	[50]
LINC01619	down	DN tissues, podocytes	miR-27a, FoxO1, ROS, CHOP, GRP78	DN	[51]
LncRIAN	down	Renal biopsy, podocyte cell	Acta2, Smad2, Smad3, miR-150	Myofibroblasts formation	[81]

## Figures and Tables

**Figure 1 ijms-22-06027-f001:**
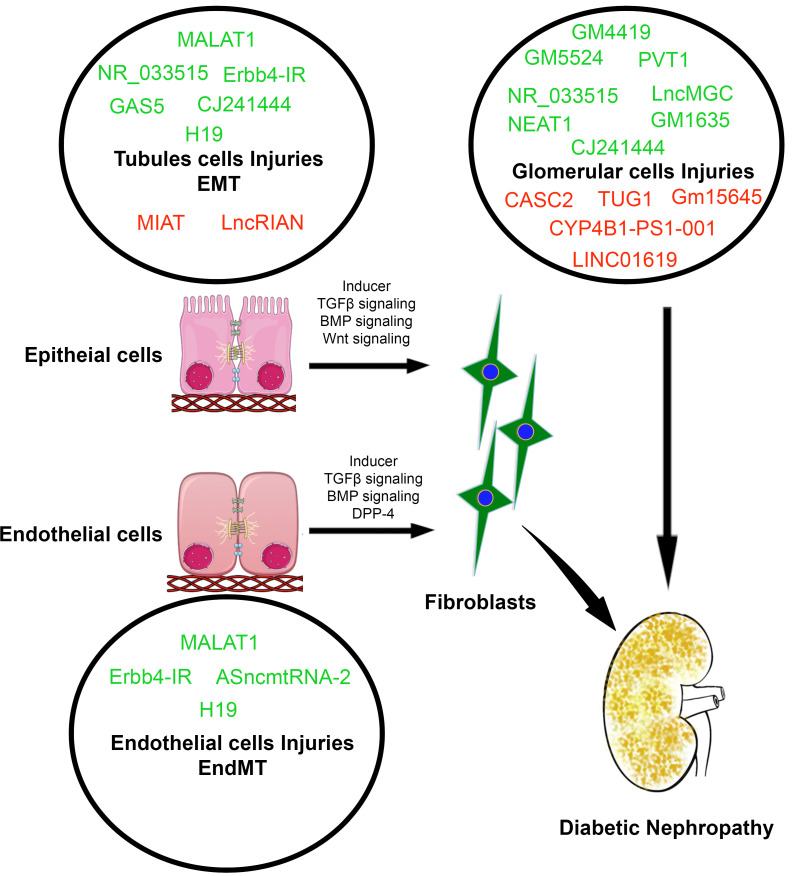
Involvement of lncRNAs in the regulation of diverse cell types in kidneys. Green color indicates upregulated expression levels whereas red color indicates downregulated expression levels during injury in cell types.

**Figure 2 ijms-22-06027-f002:**
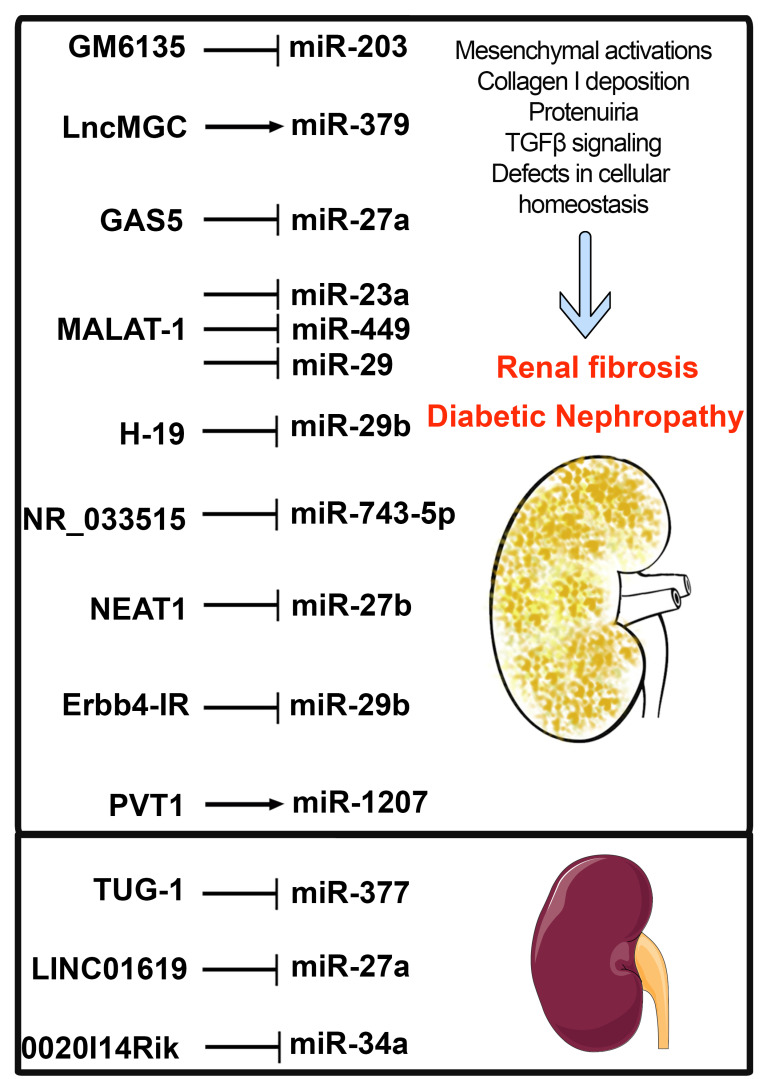
Interactions of LncRNAs and microRNAs. Arrow 
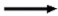

—Upregulates whereas 
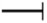
—downregulates expression level.

**Figure 3 ijms-22-06027-f003:**
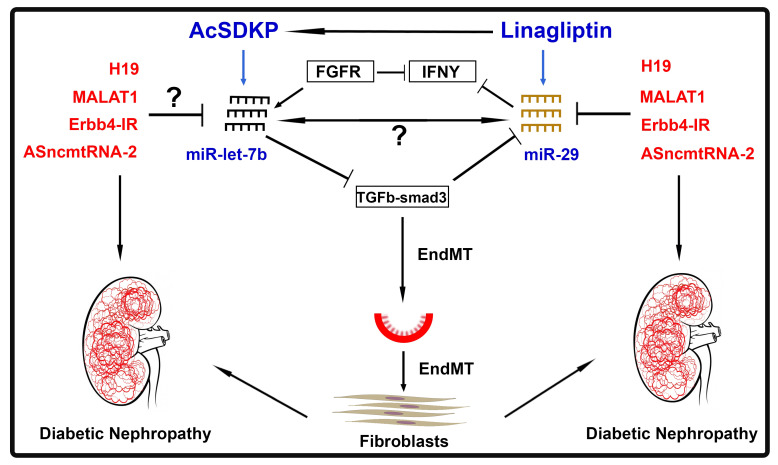
Crosstalk regulation between LncRNAs and antifibrotic microRNAs. Arrow 
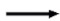
—Upregulates whereas 
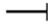
—downregulates expression level.

## Data Availability

Not applicable.

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
