# Peer review of "Interactions among Long Non-Coding RNAs and microRNAs Influence Disease Phenotype in Diabetes and Diabetic Kidney Disease"

_ijms, 2021, doi:10.3390/ijms22116027_

Round 1

Reviewer 1 Report

The review by Srivastava et al. gives a good overview of the recent publications on the involvement of lncRNAs in DN. Several minor issues require attention before publication.

1) Table 1 should include the references to all lncRNAs listed in it;

2) Some statements are unclear. For example, in lines 35-36, are the authors referring to the average numbers of exons in all protein-coding and lncRNA genes? Also, in line 214, "huge and complex" is a non-informative description. Also, in line 222, the meaning of "that are free by posttranscriptional cleavage" is unclear. In line 234 the meaning of "exert the outcomes" is unclear.

3) In general, the manuscript should be edited for style. For example, in line 257 the word "regulation" is used twice. Line 328, the word "towards" is misused. Please note these are examples, not a comprehensive list of corrections to be made.

Author Response

Reviewer 1:

The review by Srivastava et al. gives a good overview of the recent publications on the involvement of lncRNAs in DN. Several minor issues require attention before publication.

We thank the reviewer the giving us positive and suggestive comments. We have corrected it according to the suggestions.

1) Table 1 should include the references to all lncRNAs listed in it;

   We have included it in Table 1

2) Some statements are unclear. For example, in lines 35-36, are the authors referring to the average numbers of exons in all protein-coding and lncRNA genes?

We have corrected it and now it clear. Yes, we have presented a comparison of average exons between lncRNAs and mRNAs. Some non-coding RNAs do also have exons.

Also, in line 214, "huge and complex" is a non-informative description.

We have corrected it.

Also, in line 222, the meaning of "that are free by posttranscriptional cleavage" is unclear.

We have corrected it.

 In line 234 the meaning of "exert the outcomes" is unclear.

We have corrected it.

3) In general, the manuscript should be edited for style. For example, in line 257 the word "regulation" is used twice.

We have corrected it.

Line 328, the word "towards" is misused.

We have corrected it.

Please note these are examples, not a comprehensive list of corrections to be made.

We thank you for the comments. We have comprehensively revised our manuscript.

Thank you,

Best regards,

Swayam Prakash Srivastava, PhD

Yale University

United States

Reviewer 2 Report

Diabetes is one of the most important public health challenges of the 21st century. Factors underlying diabetes are complex and are not completely understood. Epigenetic mechanisms (operating via miRNAs and lncRNAs) that translate the external influences into specific genes expression provide a pivotal link between environment and genetics and therefore the lncRNAs deregulation might account for diabetes. The authors present systemic literature review of interactions between lncRNAs and miRNAs and their impact on diabetes phenotype. The paper is prepared carefully, reliably and presents important details of cited original articles, however some minor comments should be addressed to the review.

  • There is a lot of abbreviations in the text, however some of them are not explained. All abbreviations should be explained in the text or a list of abbreviations might be included. 
  • Frequently logical transition between the description of particular lncRNAs is missed. Most of the lncRNAs are described one by one without any functional connections. A reader has difficulty following the text. This issue is mainly addressed to the section 3 (LNCs in diabetic kidney disease).
  • Table 1: references related to citied data might be included
  • Table 1: formatting should be verified due to the incorrect hyphenation. The same issue is addressed to text
  • Line 94, “factor” is used twice
  • Figure 1: lncRNAs are indicated in red and green colour. Is there any reason the lncRNAs have been labelled this way.
  • Line 172: Figure 1 presents/demonstrates involvement…
  • Figure 2: Connections between left and right part of figure presented kidney is unclear. Can author include more details in the right part of figure to elucidate impact of the miRNAs-lncRNAs interactions on diabetic nephropathy.
  • Line 333-334: Excessively general description
  • Line 367-368: the phrase “problem lie” is repeated.

Author Response

Diabetes is one of the most important public health challenges of the 21st century. Factors underlying diabetes are complex and are not completely understood. Epigenetic mechanisms (operating via miRNAs and lncRNAs) that translate the external influences into specific genes expression provide a pivotal link between environment and genetics and therefore the lncRNAs deregulation might account for diabetes. The authors present systemic literature review of interactions between lncRNAs and miRNAs and their impact on diabetes phenotype. The paper is prepared carefully, reliably and presents important details of cited original articles, however some minor comments should be addressed to the review.

We thank the reviewer for giving us fruitful comments and the summary of our review manuscript. We have followed the reviewer’s suggested and made corrections to the revised manuscript.

  • There is a lot of abbreviations in the text, however some of them are not explained. All abbreviations should be explained in the text or a list of abbreviations might be included. 

We have included a new section “abbreviations” before the section “references”. The abbreviations were arranged in alphabetical order.

  • Frequently logical transition between the description of particular lncRNAs is missed. Most of the lncRNAs are described one by one without any functional connections. A reader has difficulty following the text. This issue is mainly addressed to the section 3 (LNCs in diabetic kidney disease).

We have corrected it. In the first part of section 3, we have selected LncRNAs that were upregulated in DN and described their known targets, however, in the second part we have described the LncRNAs which were downregulated in DN and their known targets and biological functions

  • Table 1: references related to citied data might be included.                   We have included it in the revised Table 1.

  • Table 1: formatting should be verified due to the incorrect hyphenation. The same issue is addressed to text

We have corrected it.

  • Line 94, “factor” is used twice

We have corrected it.

  • Figure 1: lncRNAs are indicated in red and green colour. Is there any reason the lncRNAs have been labelled this way.

We thank the reviewer for these comments. The green color indicates higher expression in injuries whereas, the red color indicates down-regulated expression levels during injuries in cell types. We have added in the figure legends.

  • Line 172: Figure 1 presents/demonstrates involvement…

We have corrected it.

  • Figure 2: Connections between left and right part of figure presented kidney is unclear. Can author include more details in the right part of figure to elucidate impact of the miRNAs-lncRNAs interactions on diabetic nephropathy.

We have included it the involvement of these interactions in the DN.

  • Line 333-334: Excessively general description

We have corrected it.

  • Line 367-368: the phrase “problem lie” is repeated.

We have corrected it.

Thank you,

With Best Regards,

Dr. Swayam Prakash Srivastava

Yale University, CT USA

Reviewer 3 Report

In this study, Srivastava SP et al. and colleagues discuss the role of Long non-coding RNAs and miRNAs influence in diabetic kidney disease.

  1. To clarify and help the readers, it would be better to add the reference numbers in Table 1 for each lncRNA target indicated.
  2. Page 5, line 150, please check the spelling for “complications”
  3. Page 6 line 172 check for the grammar of the sentence.
  4. Page 7 line 222 a hyphen “-“is missing to the word post-transcriptional
  5. Page 8, line 254 “regulating” should be changed to “regulation”
  6. Page 10, line 287, correct the spelling.
  7. Page 10, line 314, a hyphen “-“is missing.

Author Response

In this study, Srivastava SP et al. and colleagues discuss the role of Long non-coding RNAs and miRNAs influence in diabetic kidney disease.

The authors are highly thankful to the reviewer for valuable suggestions.

We have corrected and revised it according to the suggestions.

1. To clarify and help the readers, it would be better to add the reference numbers in Table 1 for each lncRNA target indicated.

We have added the references in Table 1.

2. Page 5, line 150, please check the spelling for “complications”

We have corrected the spelling.

3. Page 6 line 172 check for the grammar of the sentence.

We have corrected it.

4. Page 7 line 222 a hyphen “-“is missing to the word post-transcriptional

We have corrected the spelling.

5. Page 8, line 254 “regulating” should be changed to “regulation”

We have corrected the spelling.

6. Page 10, line 287, correct the spelling.

We have corrected the spelling.

7. Page 10, line 314, a hyphen “-“is missing.

We have corrected the spelling.

Thank you,

With Best Regards

Swayam Prakash Srivastava, PhD

Yale University, New Haven CT USA

Reviewer 4 Report

I enjoyed reviewing this manuscript. The paper is interesting and well written. The topic is hot and can grab the reader's attention.

I only have a remark on one less focused aspect in the review.

Proteinuria is a main effector of subsequent tubular damage with triggering of inflammatory and fibrotic phenomena, with alteration of tubulo-glomerular feedback, which is a central mechanism of DKD. Proteinuria determines the cardio-renal outcome of patients with DKD as has been recently documented (Nephrol Dial Transplant. 2018 Nov 1;33(11):1942-1949.  doi: 10.1093/ndt/gfy032). Actually, the authors dedicated only one sentence to this topic in the entire review (lines 164-165). I believe that this issue should be commented more widely, also citing the previous reference.

Author Response

I enjoyed reviewing this manuscript. The paper is interesting and well written. The topic is hot and can grab the reader's attention.

I only have a remark on one less focused aspect in the review.

Proteinuria is a main effector of subsequent tubular damage with triggering of inflammatory and fibrotic phenomena, with alteration of tubulo-glomerular feedback, which is a central mechanism of DKD. Proteinuria determines the cardio-renal outcome of patients with DKD as has been recently documented (Nephrol Dial Transplant. 2018 Nov 1;33(11):1942-1949.  doi: 10.1093/ndt/gfy032). Actually, the authors dedicated only one sentence to this topic in the entire review (lines 164-165). I believe that this issue should be commented more widely, also citing the previous reference.

I am thankful to the reviewer for spending valuable time reading our manuscript. In the revised manuscript we have made a significant effort in making our manuscript more focused and have corrected the manuscript based on suggestions. Indeed, we agree proteinuria is a major contributor to the development of DKD. We have reviewed and added it in the introduction section and have cited the references.

In addition, we have long-coding RNAs that regulate proteinuria levels and associated features of diabetic nephropathy.

Thank you,

Best Regards

Swayam Prakash Srivastava, PhD

Yale University, New Haven CT USA